# Enterprise Information Governance, Domain Specificity, and the Data Mesh Paradigm

Alastair McCullough*[1]*, Jim Davies*[1]*

*[1] Department of Computer Science, Oxford University, Oxford, United Kingdom*

## Abstract

This paper presents a novel critical analysis of the design of enterprise and domain specificity in information governance operating models–or target business architectures-by looking closely at the well-known data mesh sociotechnical method to understand a localized, domain specific approach. The analysis builds upon a new definition of enterprise information governance, defined as acting through control mechanisms to assure accountability in managing decision rights over information and data assets in organizations. The paper uses a graphic representation of such governance as a framework to consider the nature of strategic and tactical policies and standards that form the basis for data mesh thinking. It includes definitions of data objects and data products, and defines a technical use case, anchored in standard corporate accounting practice and software engineering, to exemplify data products. The paper then focuses on the specificity of ownership, bringing out domain, business unit, process, and decision-point bases alongside data mesh to support both design of governance and further scholarly research endeavors in consideration of domain-specific regulatory business architectures for governance.

## Keywords

Data Mesh, Domain, Enterprise Information Governance, Data Governance, Data Product, Data Object, Domain Specificity, Policies, Standards, Process, Decision-Point, Regulatory Framework, Data Strategy, Target Business Architecture, Operating Model, Software Engineering, Data Mesh Use Case.

## 1. Introduction

There is an unresolved tension hidden in information governance texts and the research and design of such governances. It lies between a definition of regulations for a whole organization and the definition of regulations localized to a department, team, or group: An individual "domain" within the organization, for example accounts payable, or sales, or software engineering, data science, or AI engineering. This paper looks at the meaning of "data products", a term which is far from theoretical because it is being introduced into a wide range of organizations today and forms the basis for software engineering works. Indeed, companies as diverse as PayPal [1], AstraZeneca [2], ABN AMRO [3], Disney [4], HSBC [5], and Michelin [6] have adopted the overarching paradigm as part of their information technology practices.

To help in understanding the relative concepts of enterprise governance, its meta-regulation and then domain-specificity, and to anchor analysis in a more real-world footing the

*NXDG, NeXt-generation Data Governance Workshop, September 3 – 5, 2025, Vienna, Austria.*

* Corresponding author.

✉ alastair.mccullough@cs.ox.ac.uk (A. McCullough); jim.davies@cs.ox.ac.uk (J. Davies)

🆔 0009-0005-1324-1975 (A. McCullough); 0000-0003-4664-6862 (J. Davies)

paper will consider a specific and still new skein of thinking. In 2019, a new sociotechnical method, or paradigm, arose and became what is today known as "data mesh." The concept and paradigm have become both eminent and significant in leading thinking across the IT industry. The paper will look at data mesh to help to understand more about domains, about some of the leading thinking today in this space, and about localized governance of information and data.

The paper will discuss regulation of data using data mesh as a governance model for specific organizational domains, and to understand more about its popular approach. The paper will present a fictional data mesh software engineering data product use case to help to look at domain specificity, drawing upon findings and the framework to help in understanding specificity and the governance of enterprise information. Research was scoped around primary data mesh sources and source materials with respect to the 2024 definition of *enterprise information governance,* upon which this paper builds directly. Methodologically, a qualitative, synthetic approach was adopted, including researcher reflexivity [7], with access to a library of materials maintained by IBM and included by relative scholarly assessment of relevance within thematic scope and a time boundary determined by research duration allocated.

## 2. Enterprise Information Governance

Enterprise information governance (EIG) is defined as encompassing both information governance and data governance [8]. It is adduced to be a corporate-wide strategic framework, differentiated from pure information technology governance and data management, developed in alignment with respect to a vision for end-to-end governance set by enterprise leadership. This vision cascades to operational levels via the defined constructs of "mission" and "goals", statements in relation to strategy for the organization of which goals are a subset of mission.

The EIG definition sees such governance as framing actions to ensure trust and compliance by determining strategic, tactical, and operational policies, standards, guidelines, and processes. Alhassan, Sammon, and Daly define governance of data and management of data as differentiated: Governance refers to decisions, whereas management involves implementation [9]. Hence, management is influenced by governance. The EIG framework itself supports management of shared resources and is delivered through rules—or other control mechanisms-that co-ordinate, integrate, direct, monitor and allocate tasks. These exercise authority, control, and accountability in order to manage decision rights over data [10].

The pursuit of EIG by an enterprise, prosecuted by an executive sponsor [11], would be undertaken to achieve value-maximization of data assets. It would need to be consistent with organizational strategy, mission, values, norms, and culture. The EIG framework is scoped and bounded through the design and definition of a business architecture. This brings together organizational leadership's strategic vision for governance, however that has been defined, with day-to-day operations, the tactical activities undertaken by individuals and teams.

As an essential aspect of the EIG definition, "information" is viewed as a superset in a technology landscape; "data" semantically a subset. This construct enables consideration of information and data as terms at relative levels within and across an enterprise. "Information" is more strategic and encompassing. Data represents operational, processable, resources:

Assets to be governed and controlled at a lower level contextually subsumed by the scope and boundary of "information" [12].

The EIG definition encompasses organizational components, business capabilities, and functions, in the form of an (initially new) *target business architecture.* Such an architecture will include definitions for a variety of classic operational governance roles, such as Data (or Information) Owner (or "data trustee"), Custodian, and Steward, and may be broadened to encompass more technical topics, including aspects of data management, security, privacy and topics that intersect with operational C-Suite [13] and divisional portfolios [14]. This perspective and definition of EIG will be used to form the basis for the consideration of the data mesh perspective with respect to *domain specificity* in the governance sphere.

## 3. Data Objects

Today, facts are held as the information and data used and stored by an organization, or upon their behalf in the case of cloud services provision, by vendors and platforms such as Google Cloud®, Amazon Web Services®, and Microsoft® Azure®. What we call "data" are typically held in operational data stores and varietal components that constitute such information technology systems and "systems of systems" [15]. Increasingly, such data are processed at scale [16] to address a variety of use cases.

Scholars have viewed data and information in various ways. For example, from the 2024 paper framing the definition of enterprise information governance and that summarized them [17]: Buckland introduced the concept of "Information as thing", processed in some manner. Madison saw "Data-as-form", both thing-like but also "simultaneously form and flow" [18]. Weber, Otto, and Österle saw data as "'raw' or simple facts" and information as "data put in a context" or processed [19]. Boisot and Canals saw "information [as] an extraction from data" [20].

In thinking about data and its governance, a useful conceptualization is then that of the "data object", defined by IBM in 2021 [21, 22] as: *"the information or data owned within the scope of ownership."* The definition introduced concepts of ownership of data, of stewardship, custodianship, and of *bounding ownership of data,* via a domain or scope definition. The concept here saw a data object as physically relating to one or more data items, themselves constituted from anything in scale from a single byte to zettabytes of data, or more. In business terms, a data object could therefore take the form of, for example, a single image file (file extensions such as .jpg, .png), a sound file (.mp3, .wav), a movie file (.mov, .mp4, .wmv), or a 100Mb document (.docx, .pdf), just as it could be a simple character-based file of 10 bytes in size (.txt, .csv), or characters representing a 35Tb binary large object (or BLOB). Such an object could live in a file system owned by a grouping with some *scope of ownership,* such as the accounts payable team, human resources team, sales team, or any other unit, group, or division. Equally, a data object could exist within another form of operational data store, a database, data warehouse, or a bucket within AWS' S3, "simple storage service". Of course, by implication a data object need not in fact be owned by a definite team, in which case ownership might be assumed to be that of "the company" in general, the organization in whose ownership the data or file system or operational data store containing the data subsists.

## 4. Data Mesh, Data Products, and Data Domains

Target business architectures are essential both to design governances and to support the transformation of organizations to encompass them. Such alterations of organizational schema are usually undertaken as part of a Change program or project, driven by a Transformation Plan [23]. Such business architectures need to make sense to individuals in teams, in departments, in projects, in scrums; in defining functional and non-functional requirements and backlogs; delivering artefacts and software, and in the activities related to, and the equitable management of, data and information.

Whilst there is clearly an enterprise, more macro, level of governance there will also exist a set of constructs and a framing for policy, standards, guidelines, and procedures, a version or element, that ensures these regulatory aspects can govern data day-to-day and proactively in relation to a lower level than the strategic and that we might appropriately call *tactical governance.* Considering data, it can be deduced that there will need to be a finer grained aspect of governance, or an extension to regulatory framing, to support the lower, tactical level of activity, whether it be a policy or standard, or a process step within a procedure.

There exists an emergent systemic approach that encompasses this finer-grained thinking. It embraces both tactical practice and a way that organizations can be transformed to consider data more particularly, looking directly at *domain and localized regulation or governance.* The method crosses the data management and data governance boundary and perhaps challenges the concept of "enterprise" information governance itself.

The method arose from a technology consulting house, ThoughtWorks, in 2019. It became what is today known as "data mesh." The concept was developed from the work of Zhamak Dehghani, then director of emerging technologies at ThoughtWorks, and was framed in their well-known introductory paper [24]. Data mesh has become both eminent and significant in leading socio-technical thinking across the IT industry, with its prominence exemplified in a paper presented by Gartner at a major international conference in London in 2023 [25].

The concept of data mesh had been developed by Dehghani in 2018 where the original work looked at why clients' investments in technology were not generating better paybacks for them. Dehghani's main conclusion was that the prevailing method of trying to force data into a single monolithic architecture, such as a data warehouse, was fundamentally limiting [26]. The new data mesh paradigm offered a way to resolve this *monolithism* through organization transformation, with a focus upon the *tactical* management of data assets.

Data mesh is most particularly a *sociotechnical* paradigm, combining the triumvirate of *people, process, and technology* [27]. A variety of IT industry practitioners, however, have understandably confused it with data fabrics [28], with which it may align, but from which it differs substantially by virtue of not being a pure technology solution, platform or necessarily an IT architecture, *per se* [29]. Opportunities for confusion remain, and one such is at video streaming company Netflix, where a diligent expert software engineering team have apparently created an elegant streaming data fabric platform, they also refer to as a "Data Mesh" [30].

ThoughtWorks' definition sees data mesh as a "paradigm shift" in data management, one that attempts to resolve the typically monolithic nature of traditional data warehousing and

data lakes.  It resonates also in organizations that strive to resolve reference data (RDM) and master data management (MDM) [31].   The methodological aspect is founded upon four principles, those of "domain-driven ownership of data", "data as a product", "self-serve data platform", and "federated computational governance" [32].  The following 2021 framing of data mesh is most concise contextually: *"…a domain-driven analytical data architecture where data is treated as a product and owned by teams that most intimately know and consume the data"* [33].

The concept of data mesh relies upon *data domain owners,* or as the method defines them, "data product owners", as responsible for determining definitions of the data products they own.  This conceptualization of "data-as-product" is similar to that developed by Buckland ("information-as-thing" [34]).  Data products also recall the IBM concept of data objects, though the differences here are significant and will be brought out later in this paper.

Dehghani says that *"Data as a product expects that the analytical data provided by the domains is treated as a product, and the consumers of that data should be treated as customers"* [35]. Further, that, *"This is a new architectural construct that autonomously delivers value. It encodes all the behavior and data needed to provide discoverable, usable, trustworthy, and secure data to its end data users."*

Data become objectified to the point where they become definitively "things"–along, in data mesh terms, with a set of attributes-and therefore governable because such parameters are *knowable, quantifiable, determinable* (according to Dehghani), and a lifecycle for them might be construed and thus theoretically controllable.  Data products have a more precise definition, in that they are seen as individuated by virtue of being required by data mesh each to conform to a coherent model: They must be secure, discoverable, addressable, understandable, trustworthy (or truthful), accessible, interoperable, and valuable [36].

Data mesh relies upon the structure of an organization being revised (or transformed) such that localized domain groups–for example, an accounts receivable team in group accounting, or a data science team in an insurance company's actuarial department-take charge of the creation and provision or delivery of *data products* in specific relation to that department. The rationale here is that what we may call such *domain specificity* ensures that interested parties are "close to" the data they use day-to-day.

The mesh model is therefore advocating decentralization of data specialists.  They become more identified with the team, group, or department in which they work rather than with any centralized function, a totem Dehghani arguably sees as anathema.  In practice, these people might be data scientists or data engineers, working as part of domain or data product teams.  Data mesh enables the delivery of customized data products to meet the demands and requirements of different interests within and across an organization.

A model use case will be framed in order to review Data Mesh and consider the data object concept.  This will be grounded in the software engineering domain to help to understand it more clearly and to consider data mesh in the context of more real-world enterprise information governance.

## 5. Using a visual model

The paper that defined EIG as a synthesis and extension of research in the field used a graphical visualization of the novel definition it had adduced. This is reproduced here as **Figure 1** to assist in reviewing data mesh and domain specificity in the next section.

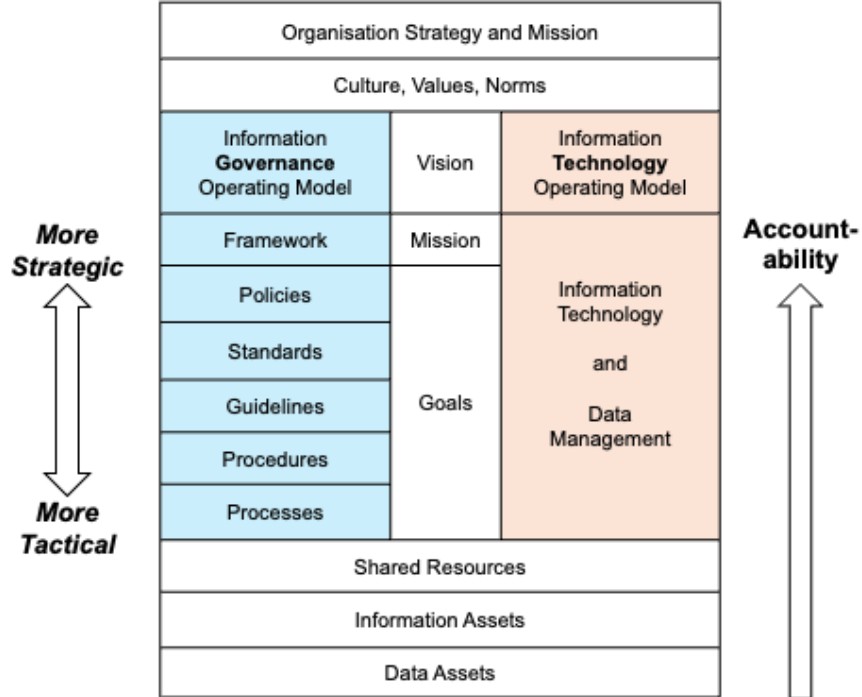

**Figure 1:** Enterprise Information Governance definition visualization graphic [37].

Towards the top of the graphic, components are more strategic in nature. For example, Organization Strategy and Mission are represented. Towards the bottom, more tactical. "Framework" represents the encompassing business architecture design (also known as an "Operating Model") for information governance and that performs as a regulatory set the exercising of "authority, control, and accountability in managing decision rights over data" [38]. It is followed by components including policies, standards, guidelines, procedures and processes that represent the regulatory building blocks of governance. "Shared Resources" in Figure 1 represents people, roles, facilities, and data resources. "Information Assets" and "Data Assets" then represent information and data within the governed organization, in which information is the meta-set, data the sub-set.

## 6. A use case model

To understand the data mesh concepts of data products and domain specificity in software engineering terms, it is useful to consider how they might appear in a particular development use case. An accounts receivable (AR) financial team case can be created to help this by analogy.

In practical terms, the Snowflake® data warehouse will function as a cloud technology platform example, as it is one widely known and used across commerce and academia today.

The Snowflake platform uses a standard Structured Query Language implementation (actually a superset of Oracle SQL, with which it is significantly compatible) which includes ANSI SQL:1999 and ANSI SQL:2003 [39], analytic extensions and database constructs such as tables, views, and materialized views.

Take the use case where the accounts receivable team are interested in understanding aged debt. This is the concept that a debtor owes money to the company when they have purchased a good or service from them but not yet settled their debt or paid the full price of their purchase. As soon as the purchase is completed, in the case either where no funds are initially exchanged or just a portion of the price is paid, such debtors enter an "aged debt" status and aged debt financial cycle, common to financial systems, and a standard construct in accountancy practice. This allows the company to understand how old a debt is, to "age" it across days and weeks, and to take action with respect to it as appropriate, for example by charging interest or referring the debt for collection or factoring.

The data mesh use case is a requirement to produce an accounts receivable aged debtor business intelligence dashboard for the AR team leader and their team members to use to help manage aged debt. The dashboard is visualized (for example) using as well-known software product, Qlik Sense® [40]. In this use case, the dashboard will access aged debtor data from a Snowflake® (data warehouse) materialized view database construct [41].

Materialized views are serverside constructs that hold only a SQL SELECT statement and settings related to the management of the SQL statement and its data. Aside from a relatively small quantity of metadata and parameters, they do not contain any other data [42]. The SQL statement is written to take data from one or more source data tables or other structures.

The materialized view looks and feels to a user or using process as a data table does, with rows and columns (tuples and domains from Relational theory [43]) containing data. It may show a snapshot of current data, or it may be refreshed with a defined periodicity, for example once per day in the morning, such that data are only ever, at worst, twenty-four hours out of date before the next refresh.

The metadata relating to the accounts receivable aged debtor business intelligence dashboard, which is effectively a BI report, will be stored in the organization's data catalogue. The catalogue will be a software tool. Example such tools include offerings from a variety of vendors at the time of writing, such as Collibra®, Talend®, Atlan™, Oracle™ Data Catalog, IBM® Knowledge Catalog, and Informatica™. The catalogue is exposed securely in the organization's IT estate such that authorized users in the business can access the information about the dashboard, potentially its component materialized view too, and understand how it can be used for dunning, modelling, or reporting upon aged debt.

For this use case, using the definition offered by data mesh the data product here would be defined as the set of (likely metamorphosing) source data and data structures, plus the materialized view, plus the report structure that utilizes the view, plus the access rights (for example, role grants and table grants) and security protocols (for example, password aging settings) particular to the operation of the view, of the report, and of the access to the data. The metadata about this dashboard are then stored in the corporate data catalogue, perhaps Collibra.

In data mesh terms, the definition of the data product is determined by matching the criteria for such an object as being secure, discoverable, addressable, understandable, trustworthy (or truthful), "natively" accessible, interoperable, and valuable. In this case, Dehghani's definitions [44] would apply.

For example, the product must be **secure** – the security protocols and parameters are defined for the data warehouse (roles, users, access), for the materialized view, for the dashboard would align here.

The product must also be **discoverable** – the information about the dashboard has been stored in a centralized registry, in this example a data catalogue, Collibra, where it will appear when sought using appropriate parameters, by a user, or potentially software. This will include details such as relevant metadata (data defining and about the product), product ownership, its origin, lineage, and versioning.

The product must be **addressable** – The product has a permanent and unique address for access via software or manually. Dehghani writes about having a "partitioning strategy and grouping of data tuples associated with a particular time (or time window)" [45] and in fact the materialized view exactly meets the addressability criterion and the concept of the window. In our imaginary use case, our combination of the Snowflake (systemic) data dictionary, the Collibra data catalogue and SQL access provide the elements that address the requirement here.

The product must be **understandable** – This is part of the data mesh concept of self-service. The concept here is that each data product provides semantically coherent data, with a specific shared meaning, therefore, that will be readily apparent to the user of the product. Ideally, data mesh proposes here that the entities that compose the data product, their relationships and adjacent data products are apparent [46]. Concepts such as "entity" and "relationship" are borne of Relational Theory and can be found concisely in data modelling methods, such as that of Barker [47].

The product must be **trustworthy** (truthful) – it assures the trustworthiness of data through defining Service Level Objectives ("SLOs") [48] with respect to the Data Product, at the same time as assuring data provenance and lineage and supporting data quality metrics. Dehghani's term, SLOs, may be recognized by service management specialists as SLAs (Service Level Agreements) and OLAs (Operational Level Agreements) [49]. In data mesh terms these actually relate to parameters such as timeliness, change interval, and operational qualities such as data "freshness."

The product must be "natively" **accessible** – A data product should be capable of its data being read ("accessed") by the standard ("native") software tooling used by a variety of users. In this use case such access is achieved via SQL, the materialized view, and via information about the data (metadata) contained in the Collibra data catalog.

The product must also be **interoperable** – It must be easy to link it across domains, that is, to other products and for uses other than the initial use case here. For example, Snowflake schemas can be linked-to and re-used; data can be linked or mapped to other products; there are shared metadata components (e.g., fields, Relational domains, or columns) and Collibra information; there is some commonality in terms of identifiers and a unique global address for the object.

Lastly, a data object must be **valuable** (on its own) – It must have inherent value in the service of the organization. Dehghani makes the point that the product *"should carry a dataset that is valuable and meaningful on its own—without being joined and correlated with other data products"* [50]. The data mesh conceptualization of a data product is therefore a particular one with which our dashboard use case conforms. It is defined as a data product in a manner that accords with data mesh as a paradigm but assumes that data mesh has been supported and introduced to service by organizational transformation such that our dashboard data product is *produced* via a mesh.

## 7. Data Mesh and Governance

The enterprise information governance framework can now be used as a toolset with which to understand the data mesh perspective, as an example of looking at *domain specificity* in the governance sphere. Data mesh in its very first, 2019, incarnation seemed from one reading to eschew classical information and data governance, as such [51]. Even today, with later updates and a 2022 book to summarize thinking, Zhamak Dehghani feels that the term 'governance', *"…evokes memories of central, rigid, authoritative decision-making systems and control processes… that become bottlenecks in serving data, using data, and ultimately getting value from data"* [52]. From a research perspective, the data mesh view of governance appears not so much to be a regulatory one, which EIG is—driven as it is by policies, standards, guidelines, and procedures, the framing of vision and mission-but more a data-centric, data mechanistic, one.

As Dehghani writes, *"The concept of a data product as an architecture quantum attempts to integrate data, code, and policy as one maintainable unit"* [53]. Governance is seen here as one of the *"manual interventions,* complex central *processes of data validation… global canonical modeling* of data with minimal support for *change,* often engaged *too late"* [54]. Dehghani feels that data mesh "inverts the model of responsibility" because data lakes and warehousing architectures centralize authority to a team, where *"Data mesh shifts this responsibility close to the source of the data,"* [55] close to a specific domain: Data mesh… *"…embraces constant change to the data landscape… and heavily automates the computational instructions that assure data is secure, compliant, of quality, and usable"* [56].

There is a conundrum here. This arises because the data mesh view of governance, of policy compliance, seems not to accord entirely with the more holistic view of EIG, which has been adduced from the work of diverse scholars researching data governance and the management of data. Susan deMaine [57], one such scholar, says that "Information governance is a holistic business approach to managing and using information." The data mesh paradigm seems to imply that it is not holistic, but localized, perhaps even "stovepiped", a concept, in one reading, from the American West in which homesteaders' wooden houses sit separately apart from one another across the wild frontier, with black iron stovepipes and chimneys delineating localized families and *separateness.* The concept of the *silo* is often used similarly in the IT industry, connoting differentiated, separated, grain siloes standing across a farm's estate.

Data mesh is driven mechanistically, more from the "shared resources" tactical aspect, with an emphasis upon stated technology models such as data lakes and data warehousing,

discursive reference to which is, in data mesh literature, legion. Governance author and researcher, Robert Seiner sees governance as the *"formal execution and enforcement of authority over the management of data"* [58]. Another recognized author, Robert Smallwood, sees it as *"processes, methods, and techniques to ensure that data at the root level is of high quality"* [59]. It would be possible to see data mesh as aligning with these perspectives.

Although data mesh seems likely to align with aspects of Relational theory in some respects, in its espousal as *"a domain-driven analytical data architecture"* (quoted earlier) the concomitant inclusion of data modelling within the cohort of data mesh thinking is not evidenced in the literature. Data modelling itself is more of a software engineering method than a governance approach – it would in fact perhaps fit most readily into the "information technology and data management" domain, to the right of the graphic in our Figure 1.

Information Governance, it can be posited [60], is itself neither primarily mechanistic nor software-centric, but *regulatory* in nature. The primary data mesh approach, by comparison, considers that data products are governed by *("computational instructions")* programmatic code that represents policies. The code is intended to be embedded in each of the data products, with each programmatic policy validated and emplaced as part of the lifecycle of the product [61]. This is a tactical, domain-bound, rather than a strategic, approach.

Thinking of the tactical nature of data mesh as we have seen it to this point, we can note that deMaine quotes Paul Tallon, a professor of information systems, in writing that *"Factors that inhibit information governance include… (2) Outdated departmental silos (IT and others) and low process integration; (3) Pack-rat mentality in the organizational culture; and (4) Decentralization"* [62]. Decentralization is a cornerstone of data mesh, but it is also a method that actively supports *departmentalism,* albeit with significant caveats.

The productized view seems at first to exclude the concept of enterprise, of *holism*, of holistic, cross-functional, cross-domain governance, though data mesh is described as requiring "business-driven execution" and being a "component of a larger data strategy" [63]. Dehghani also writes about the delivery of governance via a "mesh experience plane," [64] or "multiplane platform," [65] which could in program delivery take the form of a data platform, equating (for example) to a cloud environment, such as AWS, Azure, or Google Cloud. Again, a mechanistic approach.

Summarizing research for this paper, *explicit* assumptions that data mesh appears to make are: (i) Policies are programmatic in nature. (ii) Policies are versioned. (iii) Policies are tested. (iv) Policies are executed (i.e., run.) (v) They are time-variant and change with time.

The assumptions that appear *implicit* here, and that can be clearly inferred from data mesh literature, include that: (i) Policies are *capable* of programmatic representation and assertion (data mesh likes the idea of *"policies as code"*). (ii) Policies will be *embedded programmatically* within a data product. (iii) The embedding of a programmatic policy within a data product is not itself problematic. (iii) Policies are always necessarily mechanistic, software-centric, and programmatic in nature. (iv) Data mesh, and any governance within its scope of transformation, *relates only (realistically) to data products.* (v) Data products, to be valid, must accord with a *prescriptive scoping* rather than (say) to data objects, which might accord with a wider and less proscribed parameterization.

Dehghani proposes the concept of the "Data Product sidecar" [66, 67] an execution engine, to carry out policy execution; and the "control port", a set of interfaces for configuration, or re-configuration, of policies. The example of the notional "right to be forgotten", which arises from interpretation of the GDPR (although the Regulation does not in reality include such a right in this form [68]), is given as a *"high privilege governance function."* If we look back at Figure 1, this function is more aligned with IT and data management, than with data privacy functionally.

*Regulatory* enterprise information governance policies in themselves are not necessarily able to conform to these implicit assumptions. It may be suspected that the data mesh view of policies is more purely mechanistic for exactly this reason.

Taking for example as a use case from the field, a realistic governance standard might be one concerning data usage. This is necessarily more regulatory than mechanistic, *per se,* and a straightforward consideration as a thought experiment makes this apparent.

A thought experiment: We might consider that data usage, within the governance framework, could be a policy and a standard. An "Enterprise Corporate Data Usage" *standard* might (as an over-simplified example) include a statement that "All internally generated data to be passed outside the company for any reason must conform with data privacy regulations and no personal data such as living natural persons' names or forms of identification should be passed outside corporate systems." Intuitively, this seems plausible to define as code [69]. Let us take a case where data contain a surname and a social security number. It is relatively easy to encode within a database that neither a field called "surname", nor a field called "social security number" represented in any schema can be viewed for export, using serverside role-based obfuscation (hiding) of data, for example via Snowflake's views or materialized views or serverside or schema role assignments or a mixture of these techniques. That would be an example of a programmatic (via role granting or object creation) realization of such a data usage policy, and not in itself problematic.

However, let us suppose that the data usage *policy* includes a paragraph such as, "Users of data must consult with the Chief Information Security Office, General Counsel, and the office of the Chief Data Officer with respect to the approval of connectivity with third party data providers prior to such approval being given." This is a requirement that may rightly be included in a policy. This will accord with the explicit assumptions that data mesh expects. Considering those: (a) The policy will likely have a version date and a version number, because good document control and, incidentally, good software engineering, requires such versioning. (b) The policy could be *tested* by being read by reviewers. (c) The policy can be *executed* by being placed into service at the direction of a board or council, from a definite date. (d) The policy can be *changed* with time and a new version issued. From these perspectives, a purely *written* regulatory policy would conform to data mesh.

However, how might a development team *encode the directive on consultation?* It would require human interaction across at least three roles as framed. The method argues that *"Embedded policies are validated and imposed at the right time through a data product's life cycle, and right in the flow of data"* and gives an example of in-memory encryption of data types, and of the articulation of this policy as code, validation, and deployment. But the example does not consider non-programmatic validation or governances that might be incapable of

programmatic expression.  We can therefore see a significant divergence between the data mesh concept of domain specific "governance" policies, and by extension standards, and what we have adduced as the enterprise information governance concept of them.

Dehghani, in seeing a new data mesh paradigm in decentralized data, *"pushing ownership and accountability of the data back to the business domains,"* [70] harks back, it can be argued, to an older world than the current Cloud-dominated, multi-platform reality. Dehghani writes of organizational shifts from centralized ownership of data by specialists who run data platforms, but in many modern organizations, data is already available using localized cloud instantiations: AWS, Google Cloud, Azure, just as examples we can meet today in everyday business, wider commerce, and academia.  These instances are owned—though not necessarily paid for-by local departments, teams, or groups anyway.  The idea of a single monolithic data store for an organization most often founders on the reality that companies mostly cannot achieve such canonical master data management [71].

We can now summarize observations about data mesh as follows, using Figure 1 as a guide for the analysis:

(i) **Strategy and Mission:** Data mesh is more *tactical* in terms of operation, less strategic *("a component of a larger data strategy"* [72]).  It sees localized policies as being essential to the model, and effectively de-prioritizes or reduces focus upon enterprise policies; it espouses domain specificity with respect to data product ownership, management, and policies.

(ii) In respect of **Vision, Mission, Goals:** Whilst the introduction of data mesh to an organization is itself a strategic choice or vision, delivery is focused by domain and domain expertise and team: We saw previously that the concept of data mesh relies upon data domain owners, or as the method defines them, 'data product owners.'

(iii) **Culture, Values, Norms:** Data mesh requires organizational transformation and culture change to a model of domain-specificity: *"Distributed Domain Driven Architecture, Self-serve Platform Design, and Product Thinking with Data"* [73].

(iv) **Operating Model** (business architecture)**:** Less regulatory, more software-centric (mechanistic) in terms of programmatic policies: *"Data mesh shifts this responsibility close to the source of the data"* [74]; and *"This is a new architectural construct that autonomously delivers value"* [75].

(v) **Governance versus Technology:** More data management, *per se,* less data governance: *"domain-driven ownership of data", "data as a product", "self-serve data platform" and "federated computational governance"* [76].

(vi) **Policies or Standards:** *"It encodes all the behavior and data needed,"* and uses *"computational instructions"* [77] to do so*.*  Thus, programmatic, mechanistic, tactical, domain specific, as we have seen.

(vii) **Shared Resources:** More mechanistic, so aligning more with technology than with the governance domain, *per se: "business-driven execution"* [78].  Data mesh becomes embedded in localized teams or groups or projects, and therefore potentially with individuated governances, local to domain, and potentially local definitions of data terminology, semantics and meanings: What could be characterized perhaps as localized *data patois* (that is, relative, rather than

absolute semantics, terminology; a "patois" being a dialect, jargon or informal speech), specific to team and individual department; without care, encouraging *silos* and the reverse of *monolithism,* technological diversity.

(viii) **Information and Data Assets:** Data mesh is *data-product-focused* rather than *data-object-focused.* Data products must conform to specifics including being secure, discoverable, addressable, understandable, trustworthy, accessible, interoperable, and valuable.

## 8. Domain Specificity

We have reviewed data mesh and governance. We have considered data mesh, data products and data domains. We can now turn towards looking at *domain specificity* with some greater clarity.

Following the work of various scholars, for example, deMaine (*"information governance is holistic"* [79]*),* Seiner *("formal execution and enforcement of authority…"* [80]), and Smallwood *("processes, methods, and techniques…"* [81]), it would be possible to see data mesh as aligning with these more holistic perspectives. Likewise, in considering Boisot and Canals' information in the physical world, Buckland's information-as-thing, Weber, Otto, and Österle *("'raw' or simple facts", "data put in a context"* [82]*),* IBM's data objects, Madison's form-like data, and Dehghani's data mesh data products, these concepts seem to have some conceptual alignment.

In the corporate world, the term "governance" appears often in the management of companies and boards of directors as *corporate governance* [83]; or, more broadly, of institutions in what could be referred to as *institutional governance.* Smallwood notes that *"IG programs are driven from the top down but implemented from the bottom up,"* [84] and The Sedona Conference® (quoted by Smallwood) determines that, *"An [Information Governance] program should maintain sufficient independence from any particular department or division to ensure that decisions are made for the benefit of the overall organization"* [85]. Independence is significant, avoiding localized silos. In these cases, the conceptual link is some form of location, of denoted area, of scope, of *placement,* that defines in what way or by what means such *office, function, or power, or authority* is both determined organizationally and delimited in terms of its sphere of dominance and influence. Thinking of this, earlier this paper referred to "an unresolved tension" in governance between a definition of holistic (whole organization or cross-organization) and localized regulations.

Boris Otto's paper [86] on the morphology of data governance organizations, brings out the idea of location and scope, writing of the concept of *"locus of control"* as *"the main instance of responsibility for data governance in a company."* Otto brings out the variety of different authors' views with respect to the *"hierarchical positioning"* of the locus, for example in different functional business departments versus IT/IS department versus a shared responsibility. Otto notes that there is no clear trend across differing opinions and observes that centralized and decentralized organization is effectively a continuum. Khatry and Brown's research determines a similar concept to that of Otto, with what they call a *"locus of accountability."* They quote co-author Carol Brown in saying, *"In designing data governance,*

*the assignment of the locus of accountability for each decision domain will be somewhere on a continuum between centralized and decentralized"* [87].

Zhamak Dehghani agrees with some of the strategic and tactical thinking that we have met in our consideration of enterprise information governance and of the thinking of various well-known scholars in the field: *"The organization needs to have a top-down continuous and clear executive communication of the vision on becoming a decentralized data-driven organization, and a bottom-up enablement through technology, incentives, and education"* [88]. And, from the original 2019 paper, *"The key for an effective correlation of data across domains is following certain standards and harmonization rules. Such standardizations should belong to a global governance."* [89]. It is hard to disagree with such sentiments, and they echo common-sense practical strategic implementation and tactical *("bottom-up")* action, in the field too. Yet, as we have seen, Dehghani's data mesh thinking places operational governance firmly in the hands of practitioners localized to domain, to team, to project, to department. Data mesh relies upon the structure of an organization being transformed such that domain groups take charge of the creation and provision or delivery of *data products* in relation to that department, even as it espouses the balancing of *"local data sharing with global interoperability"* [90]. However, at the same time, the paradigm interweaves enterprise information governance and its regulatory nature with a mechanistic approach: *"Global interoperability is managed through the federated governance operating model and enabled by automated policies embedded in each and every data product"* [91].

Earlier, this paper introduced IBM's concept of the "Data Object". It noted that IBM defined terms including owner, steward and custodian and introduced the concept of *ownership of data objects* [92]. These roles are absent in data mesh, though the concept of *ownership of data products* does exist and is tied directly to domain ownership [93]. Considering data mesh, the IBM concept of ownership of data is a more nuanced one because it is *object-centric* where mesh is *product-centric.* The IBM data object is defined differently to the data mesh *product* which requires ownership of a *domain* and ownership of its *data products.* But ownership of data, of *data objects,* rather than *data products, per se,* is a driver in information governance terms according to IBM: *"In any implementation of information and data governance in the real world, one of the most challenging areas is the selection and definition of the scope of ownership of governance… The governance must mean something in the context of the business ecosystem, and it must be simple for stakeholders to understand how and within what bounds the governance operates"* [94].

The IBM paper and concept seem to align with Boris Otto's concept of *"locus of control"* and Khatri and Brown's *"locus of accountability."* Looking more closely, the IBM paper considers that there are four potential ownership scope (perhaps, *locus)* definitions: (i) Domain-based, scoped by data domain. (ii) Business Unit-based, scoped by named business or operational unit. (iii) Process-based, process-centric, bounded by a determined (named) process set, driven by function or business; and (iv) Decision-Point-Based, scoped by the components required to take "a decision", for example produce a dashboard such as that in our case study.

Aligning IBM and Otto, it might be posited, then, that Otto's locus [95] could be represented in these, being domain-based, unit-based, process-based, or decision-point-based:

Each locus offers a governance control point, but with differentiated scope boundary. The positioning of Khatri and Brown's "locus of accountability" would depend upon the way in which the operating model of the organization supports (or fails to support) enterprise information governance and ownership.

If we consider the *data object* definitions of data mesh that we met, then, against those of *data products,* it becomes apparent that a data object is defined to enable the widest possible category of data to be governed or brought within the scope of defined ownership, stewarded and in custody of a nominated custodian. By comparison, the only concept of which data mesh admits is that of the data product, an artefact which is itself defined very precisely by being required to match the criteria as secure, discoverable, addressable, understandable, trustworthy, accessible, interoperable, and valuable. This would also preclude, considering the reverse, objects or products which do not conform to such criteria.

Data mesh explains that *"…it shifts data governance from a top-down centralized operational model with human interventions to a federated model with computational policies embedded in the nodes on the mesh"* [96]. Yet, as we have seen using the simple example of an "Enterprise Corporate Data Usage" policy, the concept of enterprise, of strategy, of framework, becomes lost in the localization and the domain specificity of the Dehghani paradigm. The model focuses upon each domain's data products and upon governance-as-programmatic-policy. It assumes that many policies are effectively software-based and, as we have argued, fundamentally mechanistic in nature. If we recall our *accounts receivable aged debtor business intelligence dashboard* use case, serving to users data sourced from a materialized view, we could see that certain programmatic rules (which might indeed reasonably be called "policies") could be represented in code or in terms of data warehouse parameters or settings, and could—depending upon scope, technology and architecture-potentially be encapsulated in a data product.

In fact, data mesh does admit to having a centralized governance team as part of what the paradigm calls *"federated computational governance"*. This team is intended to establish principles, guide decision-making, and determine global versus domain governance policies. They operate during an operating model adoption phase described as "explore and bootstrap" (note the terminology here, a mechanistic one taken from traditional software operating systems.) During this phase, Dehghani says that *"Data product developers are actively establishing patterns and sensible practices. Early data product developers work collaboratively to share knowledge and learnings"* [97]. These are, of the nature of data mesh, domain centric: *"The early governance and domain teams pave the path in establishing the operating model, decision making, and policy automation. The governance function establishes the essential policies relevant to early data products and domains"* [98].

However, looking more closely, in fact the policies concept here is indeed in the main significantly, though not entirely, mechanistic: *"During this time, the majority of policies are automated to support a mesh with a large number of interoperable and secure data products"* [99]. Recalling Otto's "locus of control" and Khatri and Brown's "locus of accountability," we could use these and the IBM model of differentiated ownership scope that we met above, to determine in a more refined way what is happening with the data mesh model. We could say

that data mesh is clearly domain-based with ownership scoped by *data domain*. Dehghani writes, *"…data mesh follows the seams of organizational units"* [100]. In many instances within a complex organization, therefore, we can imagine that domain-specificity would be synonymous with Business-Unit based ownership (locus) because under data mesh, data owners are synonymous with product owners and sit within domains. The locus of control would lie within the domain and with respect to and around data products. The locus of accountability would lie with, as Dehghani says, *"people who are closest to the data"* [101]. Data mesh is clearly neither Process-based (process-centric) nor Decision-Point-Based.

However, we can recall Smallwood's observation, that governance should, "focus on breaking down traditional functional group 'siloed' approaches" [102]. Abraham, *et al.,* note that, *"Data governance specifies a cross-functional framework for managing data as a strategic enterprise asset"* [103]. DeMaine writes that, *"Top-down implementation of information governance is particularly effective at taking the holistic view of information"* [104].

## 9. Conclusion

Whilst the choice of data mesh deployment in an organization is clearly a strategic one in and of itself, it is difficult to characterize data mesh *as strategic* or, because of its particular *modus operandi,* as a paradigm that would break down silos; it might rather perform the reverse. Recalling Alhassan, Sammon, and Daly's summation [105] that governance is about decisions and management about implementation, data mesh looks decidedly like a *data management paradigm,* and far more that than a data governance one: It is a sociotechnical method with intriguing and valuable insights into data and a novel approach. It has effectively developed its own meaning; its own semantics. It represents determined, and perfectly valid, domain specificity in data governance. However, data mesh is interpreting it in another way, mechanistically and programmatically, but reliant upon a particular type of organizational transformation and upon adoption of a singular concept, the data product, with which to govern.

In considering data mesh, we may determine that where an organization has been transformed sufficiently—that is, to the point at which staff can viably operate it in a manner according with its principles-and it can be implemented (Dehghani provides a helpful readiness test [106], and earlier in this paper we quoted a number of well-known organizations as exemplars), there could yet be a significant propensity for diverse local governances, and even a localized *semantic data patois,* to supplant canonical enterprise definitions due to siloed development and domain isolation; but also likely supplanting cross-enterprise policy and standard. That localized patois is not helpful from an enterprise information perspective.

We might call this "domain specific data governance", the case where local policies, standards and procedures have been defined locally to a team, project, or department rather than being driven, defined, or directed predominantly as the policies, standards, guidelines, or procedures of enterprise information governance, which would be differentiated from it, not least by virtue of having escalation paths and enterprise decision-making and operational bodies, and an executive sponsor, all differentiated from localized versions. This would, then, exemplify domain specificity.

## Conflicts of Interest

The corresponding author has previously been an IBM Corporation staff member and is a Chartered Fellow of the British Computer Society and a Fellow of the Institution of Engineering and Technology.  The authors are unaware of any conflicts of interest.

## Acknowledgements

The corresponding author wishes very warmly to thank Dr Leon van Heerden and Linnet Sen of the IBM Consulting Data Services practice in London, and Paul Jarvis, formerly of IBM Consulting, for their support with respect to various aspects of research for this paper.

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
