# OpenReview forum: "Enterprise Information Governance, Domain Specificity, and the Data Mesh Paradigm"
_SEMANTiCS.cc/2025/Workshop/NXDG — NXDG 2025 Conditionallyed_

### Official Review · ~Rob_Brennan1 · 2025-07-23
**Obfuscates rather than clarifies but highlights a confusing area of vendor competition in the data governance space**

**Rating:** 6
**Confidence:** 4

**Review:**

This is a whitepaper-style discussion of the author's domain of "Enterprise Information Governance" applied to the industry buzzword "Data Mesh" with an example fictional use case illustrating some points.

This is not an academic-style paper. It has no methodology and relies extensively on anecdotal quotes and grey literature references. For example there is no explicit research question. It appears to be pushing an internal IBM terminology for wider adoption. It seems to be largely based on reinvigorating the debate on the Data-Information-Knowledge-Wisdom (DIKW) Pyramid
Martin Frické. 2019. The Knowledge Pyramid: the DIKW Hierarchy. Knowledge Organization : KO 46, 1 (2019), 33–46. https:
//doi.org/10.5771/0943-7444-2019-1-33 Num Pages: 33–46 Place: Wuerzburg, Germany Publisher: Nomos Verlagsgesellschaft mbH und
Co KG Section: Reviews of Concepts in KO.
Martin Frické. 2019. The Knowledge Pyramid: the DIKW Hierarchy. Knowledge Organization : KO 46, 1 (2019), 33–46. https:
//doi.org/10.5771/0943-7444-2019-1-33 Num Pages: 33–46 Place: Wuerzburg, Germany Publisher: Nomos Verlagsgesellschaft mbH und
Co KG Section: Reviews of Concepts in KO.
Xiaogang Ma, L. A. Schintler, and C. L. McNeely. 2018. Data-information-knowledge-action model. Encyclopedia of Big Data 12 (2018),
1–4. https://www.researchgate.net/profile/Xiaogang-Ma-4/publication/322465516_Data-Information-Knowledge-Action_Model/links/
Heather J. Van Meter. 2020. Revising the DIKWPyramid and the Real Relationship Between Data, Information, Knowledge and Wisdom.
Law, Technology and Humans 2, 2 (Nov. 2020), 69–80. https://doi.org/10.5204/lthj.1470 Number: 2 Publisher: Law, Technology and
Humans Journal
and many more.
However it misses the fact that for modern systems I think it is much more illuminating to define data as anything which is represented within a computing system. For example, in the European Union Data Governance Act, data is defined as "any digital representation of acts, facts or information and any compilation of such acts, facts or information, including in the form of sound, visual or audiovisual recording".
This provides a more concrete point around which to argue.
In general a discussion of regulations is absent, although they are mentioned several times.
Similarly accountability is made central but no specific models are used as far as I could see.
A certain amount of the work is reproducing work from last year's paper by the same author. This is always a balance but since the paper is already so long.

The most interesting contribution of the paper is to try and organise the competing claims of the data mesh and data fabric communities from the vendors. However no systematic basis for comparision is laid out and so this falls short from what could be a very significant study. The links to FAIR data principles could also be explored, especially in the case study.

The paper appears to be pushing internal IBM terminology, e.g. "data objects", for some of the data governance concepts discussed. A lack of awareness of competing products like Collibra, and specifically their long established data governance operating model, surfaces in places.

Overall the paper is much to long for its contribution and would benefit from summarisation. There are extensive quotes from other sources used as a basis for the text and these would benefit from presentation in tables and thematic or other analysis.

The style of English used is not clear with a lot of terminology and obscure wording. Finally the reference list is artificially extended by creating a new numerical reference every time a paper is referred to.

In the end it was not clear what the benefit was to readers for this paper, thus I find it hard to recommend it.
Nonetheless the subject matter is interesting, relevant and should make for good discussion at the workshop.

=Detailed Comments
p2 "This construct enables consideration of information and data as terms at relative levels within and across an enterprise. “Information”
is more strategic and encompassing. Data represents operational, processable, resources"

These statement may seems make sense in isolation but in practical terms it is causing further confusion about the Data-Information-Knowledge-Wisdom (DIKW) Pyramid (which has been much discussed 2 decades ago).
For example
* What use is "information" if it cannot be processed?
* Data obviously exists at every level within the organisation, not just higher ones.
For modern systems I think it is much more illuminating to define data as anything which is represented within a computing system. For example, in the European Union Data Governance Act, data is defined as "any digital representation of acts, facts or information and any compilation of such acts, facts or information, including in the form of sound, visual or audiovisual recording"

p3 "Data Object" is already a massively overloaded term, eg see NIST SP800-73, Kotlin programming language. This is not introducing clarity.

p4 "business architectures are essential both to design governances"
I think you mean governance mechanisms or controls?

p4 "Such business architectures need to make sense to individuals in teams, in departments, in projects, in scrums; in defining functional and non-functional requirements and backlogs; delivering artefacts and software, and in the activities related to, and the equitable management of, data and information."

It is not clear why you are mixing Scrum terminology in here. I think Agile is on a downward trend and not related to your main point.

p4 "Data mesh is most particularly a sociotechnical paradigm, combining the triumvirate of people, process, and technology [27].

Data Mesh mainly exists in the grey literature at this point based on the references you have provided ie it lacks peer review and is marketing-driven. It is ironic that you cite a paper to support its socio-technical basis [27] that is not about adding an information dimension to socio-technical systems, but instead dismisses  it in favour of adding "management, customer focus and innovation".
There is, as you point out an opportunity for clarifying the landscape here but that would require a principled and systematic review and that is not provided here. There is no search methodology, no defined basis of comparison, just anecdote.

p5 "Data become objectified"
Please clarify terminology. Objectified already has a meaning in English, do you mean this or do you mean converted into a "data object" in the sense of your new definition?

Fig 1: It is not clear if the RHS column should be read as the activities "IT Management" and "Data Management" or "IT" and "Data Management"
I note that the Collibra Data Governance Operating Model going back at least  decade includes everything on the left hand side.
Finally it is not clear where "Regulations" fit in, despite you mentioning them several times in the text and them being increasingly important in an EU context at least.

---

### Official Review · ~Beatriz_Esteves1 · 2025-07-25
**Reviewer Critique: "Enterprise Information Governance, Domain Specificity, and the Data Mesh Paradigm" by Georg Krog**

**Rating:** 3
**Confidence:** 5

**Review:**

**This review was provided by [Georg Krog](mailto:georg@signatu.com), Beatriz's role was only to upload it.**

Overall Assessment

This paper represents a fundamental failure in academic scholarship, characterized by unclear research objectives, methodological inadequacy, and substitution of description for analysis throughout. While the topic of enterprise-domain governance tensions is relevant and timely, the authors demonstrate profound misunderstanding of both enterprise governance operations and the analytical rigor required for meaningful scholarly contribution. The paper reads more as an extended literature review with superficial commentary rather than systematic investigation of complex organizational challenges.

Major Structural Failures

Research Design and Methodology

Complete Absence of Clear Research Questions: The introduction identifies an "unresolved tension" between enterprise-wide and domain-specific governance but never formulates this into answerable research questions. The authors oscillate between wanting to "understand more about domains" and "discuss regulation of data using data mesh" without committing to specific investigative objectives, leaving readers unclear about what the paper aims to accomplish.
Methodological Inadequacy The methodology is problematically vague and raises serious concerns about research rigor:

● "Qualitative, synthetic approach" remains undefined throughout

● "Researcher reflexivity" is mentioned without explanation

● Access to "IBM library materials" suggests unacknowledged bias

● "Time boundary determined by research duration allocated" indicates convenience sampling rather than principled selection

These methodological shortcomings fundamentally undermine confidence in the study's analytical validity.

Missing Theoretical Positioning: The paper fails to establish what specific gap in existing scholarship this research addresses or how the analysis will advance theoretical understanding beyond existing work on centralized versus decentralized governance models.

Conceptual and Analytical Failures

Fundamental Misunderstanding of Enterprise Governance Operations: Throughout the paper, the authors treat governance as abstract policy-making divorced from the operational reality of enterprise data management. They completely miss that modern governance operates through sophisticated pipeline architectures involving developers (staging) → platform (production) → compliance → data users → compliance, where stakeholders must dynamically negotiate parameters rather than simply execute predetermined rules.

Missing Collaborative Framework: The authors consistently treat governance as algorithmic process rather than ongoing collaborative capability requiring continuous stakeholder negotiation, contextual judgment, and adaptive response to changing circumstances. This represents fundamental misunderstanding of how enterprise governance actually functions at scale.

Absence of Implementation Architecture: Despite examining a paradigm being implemented across major organizations, the paper provides no framework for how governance concepts translate into operational capability through specific roles, workflows, accountability mechanisms, or measurement systems.

Section-Specific Critical Issues

Section 1: Introduction

● Fails to formulate clear, answerable research questions

● Methodology described in vague terms that provide no operational guidance

● No positioning within existing scholarly literature or identification of theoretical gaps

● Missing scope definition and contribution clarity

Section 2: Enterprise Information Governance

● Claims EIG differs from "pure information technology governance" without substantiating this distinction

● Treats governance as abstract policy-making without explaining implementation mechanisms

● Ignores technology infrastructure required to enable enterprise-scale coordination

● Fails to establish why this particular EIG definition matters for analyzing domain specificity

Section 3: Data Objects

● Treats data objects as static technical entities divorced from strategic governance workflows

● Provides circular, operationally meaningless definition ("data owned within scope of ownership")

● Completely disconnected from EIG framework established in previous section

● Ignores governance pipeline architecture essential for understanding distributed data management

Section 4: Data Mesh, Data Products, and Data Domains

● Devotes excessive space to historical narrative that adds no analytical value

● Presents Dehghani's assertions without critical scrutiny or analytical framework

● Completely absent rule-based governance framework for obligations, rights, powers, and immunities

● No analysis of how domain-specific governance integrates with enterprise requirements

Section 5: Using a Visual Model

● Introduces framework without justification or explanation of analytical purpose

● Consists entirely of describing visual elements already apparent in Figure 1

● No connection to research objectives or methodology for framework application

● Functions as academic padding that disrupts rather than advances the argument

Section 6: A Use Case Model

● Presents drastically oversimplified view treating enterprise data management as "single database query"

● Ignores sophisticated pipeline architecture characterizing real enterprise systems

● Treats data product attributes as static checkboxes rather than complex ongoing processes

● Completely missing cross-domain governance complexity and regulatory requirements

Section 7: Data Mesh and Governance

● Treats governance like "data vending machine" with predetermined authorization parameters

● Creates false opposition between "mechanistic" and "regulatory" approaches

● No analysis of stakeholder collaboration, conflict resolution, or dynamic adaptation requirements

● Missing framework for how domain policies integrate with enterprise obligations

Section 8: Domain Specificity

● Pure literature aggregation without original analysis, synthesis, or insights

● No integration of strategic leadership with operational execution through practical mechanisms

● Ignores semantic infrastructure required for distributed governance

● Provides no pathway from theoretical concepts to actual organizational capability

Section 9: Conclusion

● Makes significant claims unsupported by preceding analysis

● Introduces "localized semantic data patois" concept never developed in prior sections

● Fails to address original research objectives about enterprise-domain governance tensions

● No acknowledgment of methodological limitations or practical implications

Missing Critical Elements

Operational Reality

The paper consistently ignores how enterprise governance actually operates through:

● Strategic leadership setting frameworks and assigning roles/tasks

● Shared semantic language layers maintaining consistency across domains

● Automated translation preserving fidelity while enabling local autonomy

● Continuous validation and feedback systems detecting drift and enabling correction

● Bottom-up documentation processes where pipeline participants describe data usage

● Collaborative intelligence combining human expertise with AI augmentation

Implementation Architecture

Complete absence of practical frameworks for:

● Role-task assignment and accountability mechanisms

● Conflict resolution between competing domain interests

● Semantic consistency maintenance across organizational boundaries

● Dynamic policy adaptation based on operational experience

● Performance measurement and continuous improvement processes

Scholarly Rigor

Fundamental failures in:

● Clear research question formulation

● Systematic analytical methodology

● Evidence-based conclusion development

● Integration with existing scholarly literature

● Acknowledgment of limitations and future research directions

Recommendations

Immediate Actions Required

1. Complete Paper Reconceptualization: The current approach is fundamentally flawed and requires total revision focusing on systematic analysis rather than descriptive literature review.

2. Establish Clear Research Questions: Formulate specific, answerable questions about how data mesh addresses enterprise-domain governance tensions.

3. Develop Rigorous Methodology: Define analytical approach operationally with clear criteria for evaluation and evidence collection.

4. Add Operational Framework: Include practical implementation architecture showing how governance concepts translate into organizational capability.

5. Integrate Collaborative Governance: Replace mechanistic treatment with analysis of stakeholder negotiation, conflict resolution, and adaptive decision-making processes.

Structural Improvements

1. Eliminate Redundant Sections: Remove or completely revise Sections 5 and 8 which add no analytical value.

2. Add Critical Analysis: Replace descriptive content with systematic evaluation of data mesh claims and limitations.

3. Include Implementation Guidance: Provide practical frameworks for organizations considering data mesh adoption.

4. Connect Theory to Practice: Show how abstract governance concepts translate into operational workflows and accountability mechanisms.

5. Acknowledge Limitations: Address methodological weaknesses and their implications for finding validity.

Overall Recommendation

Major Revision Required Before Consideration for Publication

The paper in its current form does not meet minimal standards for scholarly publication. While the topic is relevant and important, the execution demonstrates fundamental misunderstanding of both the subject matter and the analytical rigor required for academic contribution. The authors appear to lack practical experience with enterprise governance systems and substitute theoretical speculation for systematic analysis.

The most serious concern is the authors' treatment of governance as abstract policy-making rather than operational capability requiring sophisticated infrastructure, stakeholder collaboration, and continuous adaptation. This conceptual failure permeates the entire analysis and renders the conclusions operationally meaningless.

Priority for Revision:

1. Complete reconceptualization focusing on operational governance reality

2. Systematic analytical framework establishment

3. Clear research question formulation and methodology definition

4. Integration of collaborative governance and implementation architecture

5. Evidence-based analysis replacing opinion-based assertions

Current Status: Inadequate for publication without fundamental revision addressing conceptual, methodological, and analytical failures throughout.

The paper has potential to contribute meaningful insights about enterprise-domain governance tensions, but this would require complete reconceptualization focusing on operational reality rather than theoretical abstraction, systematic analysis rather than literature aggregation, and practical implementation guidance rather than academic speculation.

---

### Decision · Program_Chairs · 2025-07-25

Conditionally Accepted

---

> ### Author Response · Authors · 2025-07-29
> **Withdrawal**
>
> We are very grateful to the reviewers for their comments.  We agree that the material in the paper could benefit from a more formal, more academic presentation, and the comments will be most helpful in this regard.  It will take some time for us to re-work the paper.  This may necessitate further explanation of the 'consultancy language' in which some of the source material has been cast.  We could not sensibly undertake this for NXDG25, which we plan to attend in any case, but we will certainly consider submitting the results to NXDG26.  Thank you!